# Reliability of Xsens IMU-Based Lower Extremity Joint Angles during In-Field Running

**DOI:** 10.3390/s24030871

**Published:** 2024-01-29

**Authors:** Daniel Debertin, Anna Wargel, Maurice Mohr

**Affiliations:** Department of Sport Science, University of Innsbruck, Fürstenweg 185, A-6020 Innsbruck, Austria; anna.wargel@uibk.ac.at

**Keywords:** wearable sensors, inertial measurement units, ecological validity, 3D motion analysis, gait analysis, trail running

## Abstract

The Xsens Link motion capture suit has become a popular tool in investigating 3D running kinematics based on wearable inertial measurement units outside of the laboratory. In this study, we investigated the reliability of Xsens-based lower extremity joint angles during unconstrained running on stable (asphalt) and unstable (woodchip) surfaces within and between five different testing days in a group of 17 recreational runners (8 female, 9 male). Specifically, we determined the within-day and between-day intraclass correlation coefficients (ICCs) and minimal detectable changes (MDCs) with respect to discrete ankle, knee, and hip joint angles. When comparing runs within the same day, the investigated Xsens-based joint angles generally showed good to excellent reliability (median ICCs > 0.9). Between-day reliability was generally lower than the within-day estimates: Initial hip, knee, and ankle angles in the sagittal plane showed good reliability (median ICCs > 0.88), while ankle and hip angles in the frontal plane showed only poor to moderate reliability (median ICCs 0.38–0.83). The results were largely unaffected by the surface. In conclusion, within-day adaptations in lower-extremity running kinematics can be captured with the Xsens Link system. Our data on between-day reliability suggest caution when trying to capture longitudinal adaptations, specifically for ankle and hip joint angles in the frontal plane.

## 1. Introduction

The assessment of running physiology and mechanics through wearable sensors has received strong interest from researchers of various disciplines over the last two decades. This trend mirrors the growing understanding that human movement should be studied under real-word conditions to come to valid conclusions about the underlying phenomena that will hold outside of the laboratory [1,2]. The use of wearable inertial measurement units (IMUs) to analyze running kinematics [3], dynamics [4,5], or energetics [6] in real-world settings has become a particular focus within biomechanics and motor control research. Of all available commercial systems offering IMU-based analyses of full-body kinematics, the Xsens Link system (Movella Technologies, Enschede, The Netherlands) has probably found the most frequent application in research settings [7]. The system consists of 17 IMUs that are attached to upper and lower body segments during the movement of interest, while the corresponding software estimates segment positions and relative segment orientations, i.e., joint angles, based on a calibration procedure, a linked-segment skeletal model, and a proprietary sensor fusion algorithm [8].

For example, the Xsens Link system has been applied to study how running kinematics are influenced by unstable surfaces [9], a marathon [10], performance level in trail running [11] or by the cycling-to-running transition in triathletes [12]. Although wearable systems like the Xsens Link offer potentially new insights into running physiology and mechanics, the accurate IMU-based estimation of joint movement is challenging, e.g., due to integration drift in the measured accelerations and angular velocities [13,14], soft-tissue artefacts [15], anatomical calibration errors [16], and/or disagreement between the underlying linked-segment skeletal model and physiological joint articulation [8], to name a few.

Recognizing these challenges, some studies have investigated the validity and reliability of Xsens-based full-body kinematics during walking, running, or other functional tasks [17,18,19,20,21,22,23]. In terms of concurrent validity with marker-based optical motion capture, Xsens-based joint angle measurements typically demonstrate good agreement in the sagittal plane (i.e., flexion-extension), but moderate or poor agreement in frontal plane or transverse plane joint angles [17,18,20,21,22]. Poor agreement in frontal and transverse plane motions can partially be explained by differences in the underlying biomechanical models between Xsens-based and optical motion capture, and those differences could be addressed through respective coordinate system alignments [17,21]. However, the remaining differences in joint angle trajectories are due to technological errors and may be particularly large for ankle joint movements [17].

The reliability of Xsens-based joint angle measurements, specifically for running, is much less well understood. Al-Amri and colleagues [18] investigated the between-day reliability of Xsens-based joint angles during walking, squatting, and jumping. Similar to the pattern of validation studies, between-day reliability was generally high (Intraclass Correlation Coefficients, ICCs > 0.8) with respect to sagittal plane joint angles, while frontal and transverse plane joint angles showed many instances of poor reliability, e.g., ankle eversion during walking (ICC < 0.5). To our knowledge, the only study to investigate the reliability of Xsens-based joint angles during running was a follow-up study by Trott and Al-Amri [19], albeit only published in abstract form. They confirmed good between-day reliability in knee and hip flexion–extension range of motion during treadmill running (ICCs > 0.8), but reported poor between-day reliability of sagittal plane ankle range of motion (ICC < 0.6 for left ankle) [19]. While these two studies provide important insight into the reliability of Xsens-based joint kinematics during walking and running, many open questions remain. First, the between-day reliability of Xsens-based joint angles during running in planes other than the sagittal plane is unknown. This is problematic given that frontal plane joint angles such as hip ab-/adduction and ankle eversion/inversion are frequently discussed in running-related literature, particularly in the field of injury prevention [24]. Second, the reliability of Xsens-based joint angles during over-ground running at a self-selected speed is unknown, which is an important aspect, since IMU-based motion analysis systems are meant to be used outside of the laboratory, where running speed is typically unconstrained. Third, the existing studies determined the between-day reliability of Xsens-based joint angles based on only two testing days, which may not be sufficient to accurately estimate reliability [25]. Finally, and similar to previous validation studies [17,21], it may be valuable to investigate the source of between-day variations in Xsens-based joint angle measurements so that future longitudinal studies, which rely on repeated measurements of running kinematics, can be designed in such a way as to minimize between-day errors.

The specific research objectives of this study were:To determine the between-day, within-day and calibration reliability of discrete hip, knee, and ankle joint angles in the sagittal and frontal planes as quantified by the Xsens Link system during running at a self-selected speed on a stable asphalt surface and an unstable woodchip surface based on more than two measurement sessions each;To investigate potential sources of between-day variations in Xsens-based discrete hip, knee, and ankle joint angles by determining the association of between-day variations in discrete joint angles with between-day variations in running speed and stride frequency, as well as with different running surfaces.

## 2. Materials and Methods

### 2.1. Participants and Study Design

A group of 17 recreational runners (8 female, 9 male) volunteered to participate in five running sessions on five separate days for this reliability study. Inclusion criteria were (1) a minimum of one running session per week for at least one year and (2) no disruption of running for more than two weeks in the last six months (e.g., due to lower extremity injury or general overload). This study presents a secondary analysis of data from a previous investigation on the influence of running surface stability on whole-body running kinematics [9]. For this reason, the sample size was guided by an a priori power analysis based on expected effect sizes with respect to the running surface. Nevertheless, we conducted a post-hoc sensitivity analysis for our group of runners based on recommendations regarding how to estimate the sample size for reliability studies [26,27]. Specifically, with a group of *n* = 17 runners and *k* = 2 repeated measurements (i.e., comparison of two runs within the same day or two separate testing days), we were able to estimate ICCs with an expected magnitude of 0.8 (taken from [19] as an estimate for the true ICC) and with the minimum precision set to 0.18. This was an acceptable precision to distinguish poor reliability (ICC < 0.5) from good (ICC > 0.75) and excellent reliability (ICC > 0.9) [28].

The study was conducted in accordance with the Declaration of Helsinki and approved by the local Ethics Committee at the Department of Sport Science, Universität Innsbruck (ethics approval ID 39/2020), and all participants provided written informed consent prior to inclusion in the study. Each participant undertook a series of five testing sessions with inter-session intervals averaging 1.9 ± 1.2 days (range 1–6 days). Each testing session encompassed a 10 min warm-up period with five minutes dedicated to warming up on an asphalt surface and five minutes on a woodchip track surface (see Figure 1 in [9]). Subsequently, kinematic measurements were conducted on each of these surfaces. The sequence in which the surfaces were presented to the participants was balanced–randomized but was held constant across the five testing sessions for each individual. The duration of the warm-up phase served a dual purpose: firstly, to acquaint participants with running while wearing a motion capture suit, and secondly, to establish their preferred running speed, thereby mitigating subsequent alterations in speed during the measurement phase. Participants completed all testing sessions with the same personal running shoes.

### 2.2. Experimental Protocol

The detailed experimental protocol has been described elsewhere [9]. Briefly, for each testing session, runners wore an Xsens Link suit (Movella Technologies, Enschede, The Netherlands), which includes 17 IMUs distributed on prescribed body segments according to the Xsens Link manual and pre-defined sensor positions for each segment via Velcro stripes and pouches within the suit. The suit sizes varied among participants based on their anthropometry, ensuring a skin-tight fit and comparable relative sensor positions. The Xsens Link suit sampled IMU data at 240 Hz in the “on-body recording mode” such that running IMU data were stored on a data logger within the suit, while data processing was done after the experiment. Following the warm-up, runners completed four runs along a 140 m straight track per surface, leading to a total running distance of 560 m per day and surface. Specifically, runners completed the first 140 m in one direction (run 1), turned around and immediately completed the second 140 m in the other direction (run 2). Following a rest period of 2–3 min (save data, restate instructions, restart data logger), runners completed runs 3 and 4 analogous to runs 1 and 2. To exclude the possible effects of the turn on our reliability estimates, the subsequent analysis only focused on runs 1 and 3 (see Figure 1). Before each new surface, participants performed two separate walking calibration trials according to the manufacturer guidelines. This calibration trial consists of starting in a static standing position (the “N-Pose”), walking forward for a few seconds, turning, and then returning to the initial standing position. During the running trials, participants were instructed to run at a self-selected but constant running speed that they would choose for a 60-minute moderate intensity run. Timing gates in the middle section of the 140 m-long track were used to measure the average running speed (Figure 1).

### 2.3. Data Processing

The Xsens MVN Analyze software (v. 2021.0.1) was used to post-process all running measurements in the “High-definition reprocessing” mode and the “no-level” processing scenario. In this scenario, the position of the pelvis segment is fixed in space and all kinematic quantities are expressed relative to the pelvis, which is the recommended scenario for joint angle analysis in biomechanics [22]. During this step, the Xsens MVN software also determines 3D joint angles for all joints included in the Xsens biomechanical model, which follows the recommendations of the International Society of Biomechanics for defining coordinate systems for the ankle, knee, and hip [29,30]. All further processing steps were conducted in a custom-written Matlab script (The MathWorks Inc. (2023). MATLAB version: 9.14.0 (R2023a), Natick, MA, USA). Individual running gait cycles were identified and segmented based on maxima in the anterior–posterior position of the right foot segment, as described previously [9]. Although this method leads to gait cycles that start slightly before the actual right foot contact, it yields a reliable segmentation approach that should not introduce further error into the analyzed discrete joint angles. Although this method leads to gait cycles that start slightly before the actual right foot contact, we assume its simplicity in yielding a more reliable segmentation approach compared to previous detection methods based on acceleration thresholds. The latter may be prone to false detections in the presence of variable running patterns, speeds, and surfaces, and could therefore bias our reliability comparisons [31]. All joint angle trajectories were then time-normalized, such that each gait cycle consisted of 101 data points with the first and last data point representing subsequent maxima in the right foot anterior–posterior position. We extracted about 40 gait cycles for each run depending on the runners’ selected speed while omitting gait cycles during the acceleration and deceleration phases. We then determined one average, time-normalized angle waveform for each investigated joint, calibration–run combination, surface, and testing day for each participant for further analysis. For the purpose of this study, we focused the analysis on initial hip flexion–extension (IHF), initial and peak hip abduction–adduction (IHA, PHA), initial and peak knee flexion–extension (IKF, PKF), initial and peak ankle dorsiflexion–plantarflexion (IAD, PAD) and initial and peak ankle eversion–inversion (IAI, PAE). Initial values were taken at 0% of gait cycle and peak values during the approximated stance phase (0–40% of gait cycle) (see Appendix A for an illustration of the variables). We also determined the average running speed (RS) based on the timing gate data and average stride frequency (SF) based on the duration of each of the detected gait cycles.

### 2.4. Reliability Analysis

To address our first objective (I), we assessed the reliability of each investigated discrete joint angle based on three different comparison types:Between-day comparisons were based on the variation in discrete joint angles between the first runs on each of the five separate days on the same surface (e.g., run 1 on five separate days on the woodchip track, see Figure 1). Specifically, we based reliability estimates on a total of five day-to-day comparisons—Day 1 vs. 2, Day 2 vs. 3, Day 3 vs. 4, Day 4 vs. 5, and Day 5 vs. 2. Day-to-day comparisons were carried out pairwise to enable the comparison of between-day vs. within-day reliability, which also only relied on two measurements, i.e., two runs within a day. While between-day reliability could have been estimated using all five testing days, the resulting confidence intervals of an ICC based on five measurements would have been systematically more narrow compared to ICC estimates based on two measurements, and thus would have hindered a fair comparison of between-day vs. within-day reliability estimates and interpretations. Day 5 was compared to Day 2 to have a fifth comparison pair, equalizing the number of comparisons underlying the median ICCs;Within-day comparisons were based on the variation in discrete joint angles between two separate runs on the same surface and day, and processed with the same calibration trial (e.g., run 1 vs. run 3 on the woodchip track on day 1, see Figure 1). Given the five testing days, we based reliability estimates on a total of five within-day comparisons;Calibration comparison was based on the variation in discrete joint angles between two copies of the same run, but processed with different calibration trials (e.g., run 1 processed with calibration 1 vs. 2 on the woodchip track on day 1, see Figure 1). Given the five testing days, we based reliability estimates on a total of five calibration comparisons.

We initially summarized the absolute between-day differences in each investigated discrete joint angle (between-day error in Figure 1) along with the corresponding absolute within-day differences in those joint angles due to different runs (within-day error in Figure 1), or due to different calibrations (calibration error in Figure 1). For a given day-to-day comparison (e.g., 1-to-2), the within-day differences were determined based on the first of the two days (day 1 for 1-to-2). Then, we determined the average and standard deviation of the absolute between-day, within-day, and calibration-related differences across all five day-to-day comparisons and participants.

The between-day reliability, within-day reliability and calibration reliability of discrete hip, knee, and ankle joint angles were determined according to the intra-class correlation coefficient (ICC) and the minimal detectable change (MDC). The ICC was calculated to represent the degree of absolute agreement between measurements (“case 2” model ICC(A,1)) as defined by McGraw and Wong [32]. The MDC was estimated based on the standard error of measurement (SEM) following the recommendations by Atkinson and Nevill [25]:(1)$$\mathrm{SEM}=\sqrt{\mathrm{MSE}},$$
(2)$$\mathrm{MDC}=1.96\;\cdot \mathrm{SEM}\;\cdot \;\sqrt{2},$$
where MSE is the mean squared error of the repeated measures analysis of variance underlying the calculation of the ICC. The reported ICCs and MDCs for between-day, within-day and calibration reliability are the median and range of ICCs across the five testing days (or day-to-day comparisons). ICCs were interpreted according to the guidelines set out by Koo and Li [28] such that cut-off values of ICC < 0.5, 0.5 ≤ ICC < 0.75, 0.75 ≤ ICC < 0.9, and ICC ≥ 0.9 were categorized and interpreted as showing poor, moderate, good, and excellent reliability. The ICC interpretation could encompass two categories if the minimum ICC fell into a different category than the median ICC across days. In these cases, we interpreted conservatively, e.g., an ICC [range] of 0.95 [0.76, 0.99] was interpreted as good instead of excellent reliability.

To address our second objective (II), we calculated the between-day reliability of running speed and stride frequency according to the aforementioned comparison types. We further curated a “difference matrix” containing the absolute differences in each of the investigated discrete joint angle outcomes for five different day-to-day comparisons (1-to-2, 2-to-3, 3-to-4, 4-to-5, 5-to-2) and two surfaces (asphalt and woodchip), leading to 10 entries for each participant. In parallel, we determined the corresponding absolute between-day differences in running speed and stride frequency. This procedure resulted in a 170 rows × 4 columns error matrix per investigated discrete joint angle (17 participants × 5 day-to-day comparisons × 2 surfaces = 170 rows × 4 columns (difference in joint angle, speed, stride frequency, and the surface)). Then, we analyzed whether the running surface and/or between-day differences in speed and stride frequency were significant predictors of between-day differences in Xsens-based joint angles using one linear mixed model for each investigated joint angle variable. The linear mixed model computations were conducted in jamovi software (v. 2.3.21) [33] using the *GAML*j module with its default settings [34] (version 2.6.6). Within each investigated model, running speed, stride frequency and running surface were defined as fixed effects ($\bar{b}$), while the intercept within each runner cluster (aj) was defined as a random effect reflected by the model Equation (3):(3)$${\hat{y}}_{cj}=\bar{a}+\left(a_{j}-\bar{a}\right)+{\bar{b}}_{Speed}\cdot x_{speed,\;ij}+{\bar{b}}_{SF}\cdot x_{SF,\;cj}+{\bar{b}}_{Surface,cj}\cdot x_{Surface,\;cj}+\epsilon_{cj}$$
where ${\hat{y}}_{cj}$ is the predicted absolute between-day difference in a given joint angle for a given runner-cluster *j* and a given day-to-day comparison *c,*
$\;\bar{a}$ and $a_{j}$ are the average and runner-dependent intercepts, $\bar{b}$ is the average coefficient across runners, and ϵ is an error term representing unexplained variance.

## 3. Results

### 3.1. Runner Characteristics

The group of investigated runners had an average age (±SD) of 26 ± 2 years. Female runners (*n* = 8) were 1.68 ± 0.07 cm tall with a body mass of 60 ± 7 kg. Male runners (*n* = 9) were 1.82 ± 0.03 cm tall with a body mass of 73 ± 7 kg. On average, runners in this sample completed 2.6 ± 1.4 running sessions per week with a session duration of 58 ± 19 min and a weekly running distance of 29 ± 23 km. Such runners can be classified as recreational runners [35].

### 3.2. Between-Day, Within-Day and Calibration Reliability of Xsens-Based Lower Extremity Joint Angles

The average magnitude of between-day variations in discrete joint angles was generally larger—about twice as high—compared to the corresponding magnitude of within-day variations due to either separate runs or different calibration files (BD vs. WD and BD vs. Cal in Figure 2). This was true for all analyzed joints and both surfaces. No clear trend emerged when comparing the magnitude of within-day variations in discrete joint angles between separate runs (WD), or from the same run but based on different calibration files (Cal). The one exception was the frontal plane ankle angle (IAI, PAE in Figure 2), for which within-day differences due to calibration were more pronounced compared to the differences between runs. Similarly, no clear trend emerged when comparing the magnitude of joint angle variations between running surfaces. Again, the frontal plane ankle angles were the exception, with more pronounced calibration-related differences for runs on the woodchip track compared to runs on asphalt.

The ICC medians and ranges (min and max ICCs) based on five comparisons for between-day, within-day and calibration reliability are summarized in Table 1, split by variable category (joint angles for the hip, knee and ankle as well as spatio-temporal parameters), split by running surface (asphalt and woodchip), and color-coded according to their reliability category (poor, moderate, good, excellent). With regard to the hip joint, hip flexion (IHF) showed good between-day reliability, while hip add-/abduction showed poor (IHA) to moderate (PHA) between-day reliability for both surfaces. Within-day reliability and calibration reliability were excellent for hip flexion (IHF) and predominantly good for hip add-/abduction (IHA and PHA). Discrete knee joint angles in the sagittal plane (IKF and PKF) revealed good to excellent between-day reliability, as well as excellent within-day and calibration reliability for both surfaces. With respect to the ankle, between-day reliability was moderate to good for initial ankle angles (IAD and IAI), whereas peak ankle angles (PAD and PAE) showed poor between-day reliability. For within-day reliability, all ankle angles showed excellent median ICCs (>0.9); however, some minimum ICCs were poor (PAD on asphalt) to moderate (IAD on asphalt, PAD on woodchip), leading to low reliability categories for these angles. Similarly, all ankle angles showed good to excellent calibration reliability with the exception of PAD and IAI on the woodchip track (moderate reliability).

A closer look into the median MDCs (Figure 3) confirms the lower reliability, i.e., larger MDCs, of comparisons of runs between days vs. within days. For within-day comparisons, the MDCs range from a minimum of 1.30° for peak hip adduction on asphalt to a maximum of 2.95° for initial knee flexion on the woodchip track. For between-day comparisons, MDCs range from a minimum of 3.17° for peak knee flexion on the woodchip track to a maximum of 8.22° for peak ankle dorsiflexion on asphalt. There were only two variables where the median MDC was smaller than or equal to the average change of the respective joint angle between the two running surfaces: (1) Within-day comparisons of peak hip adduction (median MDC of 1.30–1.38° with an average surface difference of 1.42°) and (2) within-day and between-day comparisons of initial knee flexion (median MDC of 2.38–2.95° (within-day) and median MDC of 4.60–4.73° (between-day) with an average surface difference of 4.65°).

### 3.3. Potential Sources of Between-Day Variations in Xsens-Based Lower Extremity Joint Angles

In addition to the measurement technology, we investigated variations in the participants’ running styles (expressed through running speed and stride frequency) and the running surface as potential explanatory factors for the magnitude of between-day variations in joint angle outcomes. Table 1 indicates that running speed showed good between-day reliability. The mixed model analyses further showed that between-day differences in running speed were a significant predictor for between-day differences in initial hip flexion, and explained 26% of the respective variance (Table 2). On average, an absolute between-day difference in running speed of 1 m/s was associated with an absolute between-day difference in initial hip flexion of 4.23°. Stride frequency showed good reliability on asphalt but only moderate reliability on the woodchip track (Table 1). According to the mixed model analyses, 6% (peak knee flexion) and 12% (peak ankle dorsiflexion and eversion) of the absolute between-day differences can be explained through differences in stride frequency (Table 2). There was no significant association between the running surface and any of the analyzed between-day joint angle differences.

## 4. Discussion

The objective of this study was to investigate the between-day and within-day reliability of Xsens-based lower extremity joint angles during running at a self-selected speed in the field while considering a range of potential sources for between-day variations in joint angle outcomes, including system calibration, variation in running speed and stride frequency, and the running surface.

### 4.1. Within-Day Reliability of Xsens-Based Lower Extremity Joint Angles

When comparing two subsequent runs on the same day, initial and peak hip and knee joint angles generally showed good to excellent reliability with high ICCs and relatively low MDCs. For peak hip adduction and initial knee flexion, the MDCs were smaller than the average change in these joint angles when running on asphalt vs. the less stable woodchip track. A similar trend was observed for the initial hip flexion angle. Taken together, these results can be interpreted to show that the IMU-based Xsens Link system has sufficient reliability and is well suited to detect within-day adaptations in sagittal knee and sagittal/frontal hip joint movement in response to different running surfaces or comparable interventions, such as variable footwear. Importantly, this statement only applies to study designs that do not require a re-calibration between experimental conditions given the added measurement error of the calibration procedure.

The median within-day ICCs for ankle angles also pointed towards good to excellent within-day reliability, but showed some instances of only moderate or poor reliability in the sagittal plane (e.g., minimum ICC of 0.30 for peak ankle dorsiflexion in Table 1). These instances of lower reliability cannot be explained by calibration issues given that the same calibration trial was used to process the two contrasted runs. Therefore, we assume that instances of low within-day reliability in ankle angles may result from undesired relative movements between the foot IMU sensor and the running shoe throughout a running trial, and thus an invalid calibration of the sensor-to-segment orientation. Future investigators should ensure a reliable method of fastening the foot IMU sensor and avoiding undesired sensor movement for any surface or running speed setting, e.g., by securing the sensor with additional tape instead of the Velcro-based attachment provided by the manufacturer.

### 4.2. Between-Day and Calibration Reliability of Xsens-Based Lower Extremity Joint Angles

The between-day reliability was clearly lower than the within-day reliability for all investigated joint angles, as demonstrated by the lower ICCs, higher MDCs, and higher absolute between-run differences. Nevertheless, good to excellent between-day reliability was observed for initial hip flexion, initial and peak knee flexion, and ankle dorsiflexion. In parallel, these angles showed excellent calibration reliability. For initial knee flexion in particular, the between-day MDC was still smaller than the average change in knee flexion between the asphalt and woodchip surfaces. This suggests that Xsens-based running analyses may be reliable enough to monitor longitudinal changes in knee flexion angles and potentially initial hip and ankle sagittal angles over time, provided those changes have a magnitude of at least 4°.

In contrast, between-day reliability was lower for initial hip add-/abduction and ankle inversion (moderate reliability), and was poor for peak hip adduction, peak ankle dorsiflexion, and peak ankle eversion. This finding agrees with previous reliability studies of Xsens-based motion capture showing low reliability for frontal plane joint angles during gait in general [18], with a particularly high measurement error for the ankle joint [17,19]. Consistent with this finding is the observation that peak ankle dorsiflexion and initial ankle inversion were the only variables to show only moderate calibration reliability. Robert-Lachaine and colleagues [36] estimated the reliability of the N-Pose that is used as part of the Xsens calibration procedure and that defines the neutral segment orientations [22]. They showed that the orientation of the ankle’s axes of rotation can vary by up to 3° following repeated calibrations, which could contribute to poor between-day reliability. However, for peak ankle eversion and peak hip adduction, we observed poor between-day reliability despite good calibration reliability. Two possible explanations for this disconnect are: First, our study design only allowed us to estimate within-day calibration reliability, i.e., based on two calibrations on the same day. It is possible that runners’ calibration movements deviated significantly more between calibrations on different days. Second, between-day differences in IMU placement on the thigh, shank, and shoe may lead to a randomly varying influence of soft-tissue and shoe deformation on peak frontal plane hip and ankle angles during the stance phase of running. This would be an additional source of between-day errors independent of the calibration [15].

The moderate to poor between-day reliability and high MDCs for ankle and hip angles in the frontal plane are of particular concern given that (1) these angles are often discussed in the context of running injuries [24], and (2) these angles have a small range of motion. Therefore, for the example of peak ankle eversion, the MDC of 6.7° already covers more than 30% of the ankle inversion–eversion range of motion during the gait cycle (cf. Appendix A). In summary, it appears difficult to reliably track hip and ankle angles in the frontal plane with the IMU-based Xsens system across different days. Based on the results of this study, Xsens-based measurements can currently not be recommended for monitoring longitudinal changes in frontal plane hip and ankle angles during running. One idea to potentially improve the between-day reliability in frontal plane joint angles would be to monitor the exact N-Pose used by runners on the first measurement day (e.g., based on goniometric measurements, foot prints, or photographs), and then help runners to attain an almost identical N-Pose on following measurement days [36].

### 4.3. Potential Sources of Between-Day Variations in Xsens-Based Lower Extremity Joint Angles

The absolute between-day differences of the investigated Xsens-based lower extremity joint angles provide an overview of their absolute reliability in the original unit of measurement (see Figure 2). This analysis has revealed that discrepancies in discrete joint angles between measurements on two different days are about twice as large as discrepancies between measurements on the same day, confirming the trends observed in the ICCs and MDCs. Further, the analysis showed that within-day variations of joint angles have a similar magnitude for comparisons of (1) two separate runs processed using the same calibration trial and (2) the same run processed with two different calibration trials. The first comparison reveals variations in joint angles due to (1a) runner-internal processes, such as the ability to reproduce the same movement pattern under comparable external conditions, and (1b) measurement errors at the level of the IMU sensors, the Xsens sensor fusion algorithm, or changes in the sensor-to-segment alignment following undesired sensor movements (see Section 4.1). The second comparison reveals variations in joint angles due to (2a) the runner-internal ability to perform repeated calibration movements in a similar way, and/or (2b) measurement errors at the level of the Xsens calibration algorithm.

In conjunction, the observations in Figure 2 suggest that—when running at a self-selected speed outdoors—about half of the between-day differences in Xsens-based joint angles may be explained by day-to-day variability internal to the runner, while the other half may result from technical errors within the measurement system, including sensor placement, calibration, and IMU measurement errors. Additional error sources that were not accounted for in this study include environmental factors such as temperature or precipitation [37]. The fact that the movement pattern of runners varies from day to day is not new; e.g., Benson and colleagues showed that at least five self-paced outdoor running sessions on different days are necessary to fully characterize a runner’s movement pattern [38]. Our study adds that when the running movement is quantified on different days using the Xsens Link suit, the between-day variations will include an additional technology-based source of variation that is of similar magnitude to the internal source of variation.

### 4.4. Comparison with the Reliability of Other Assessments of 3D Running Kinematics

Besides the gold standard of marker-based optical motion capture (OMC), IMU-based methods and markerless (computer vision-based) methods have become viable alternatives for the assessment of 3D running kinematics over the last two decades. Therefore, it was of interest to contrast the reliability estimates between the IMU-based Xsens system presented here with previous reliability studies investigating the gold standard or emerging markerless motion analysis techniques. Table 3 shows comparisons of between-day ICCs of the current study with those from two previous studies reporting the between-day reliability of discrete lower-extremity joint angles during running at a self-selected speed based on OMC [39,40], and with one study based on markerless motion analysis [41]. In general, markerless motion analysis showed the same (three out of nine variables) or superior (six out of nine variables) between-day reliability category compared to the Xsens-based approach of the current study. This trend towards the improved reliability of markerless motion analysis likely stems from the minimal dependence of this technology on user input with respect to sensor placement or subject-specific calibration procedures [42]. When comparing the Xsens-based and OMC-based approaches, superior reliability depended on the specific variable and the marker model used in the OMC-based approach. All three technologies showed good to excellent between-day reliability (ICCs > 0.87) for initial ankle dorsiflexion, indicating that all three motion analysis approaches are suited to monitoring day-to-day changes in the foot strike angle. Knee and hip flexion outcomes also showed good between-day reliability for all three technologies, with the exception of the OMC approach, when using the Plug-in-Gait marker model, resulting in only poor to moderate reliability. The partially inferior reliability of the OMC approach combined with the Plug-in-Gait marker model may result from the strong dependence of this technique on reliable marker placement [39]. While an in-depth analysis of reliability differences between systems was out of the scope of this manuscript, Table 3 can be used as guidance when trying to select the most reliable measurement system for a given discrete joint angle during running.

### 4.5. Limitations

We did not conduct a systematic between-rater comparison, and thus our results should technically not be generalized to other raters. However, a previous between-rater reliability study on squatting, jumping and walking movements concluded that Xsens-based joint angles are not influenced by user expertise [18]. This is not surprising because the only real influence of the user on the measurement procedure results from the placement of the IMU sensors, which is predetermined and guided by Velcro straps inside the Xsens Link suit and video tutorials of the manufacturer. Therefore, we assume that the influence of the rater is negligible for Xsens-based joint angle measurements relative to the influence of re-calibrating the suit, and further, that the reported within- and between-day reliability estimates can be used for the study design and sample size estimations of future running-related motion analysis studies.

## 5. Conclusions

During running at a self-selected speed outdoors, the IMU-based Xsens Link suit derives estimates of lower-extremity joint angles in the sagittal and frontal planes that generally show good to excellent reliability between repeated runs on the same day. Provided the sensors do not require a re-calibration between experimental conditions, the Xsens Link suit is well suited to capturing within-day adaptations in the movement pattern of the lower extremities in response to different running surfaces or similar interventions. For repeated measurements on different days, the Xsens Link suit retained good to excellent between-day reliability for sagittal plane hip, knee, and ankle angles just before foot contact, and can be used to reliably monitor longitudinal changes in these angles if these changes exceed 4°. Hip and ankle frontal plane angles showed only poor to moderate between-day reliability, likely due to a higher calibration error, and thus, the Xsens Link suit can currently not be recommended for use in monitoring day-to-day changes in those variables. Potential ways to improve between-day reliability include (1) controlling running speed (specifically for initial hip flexion), (2) controlling stride frequency (specifically for peak knee and ankle angles), and (3) standardizing the subject-specific calibration pose (N-Pose) between testing days (specifically for frontal plane angles).

## Figures and Tables

**Figure 1 sensors-24-00871-f001:**
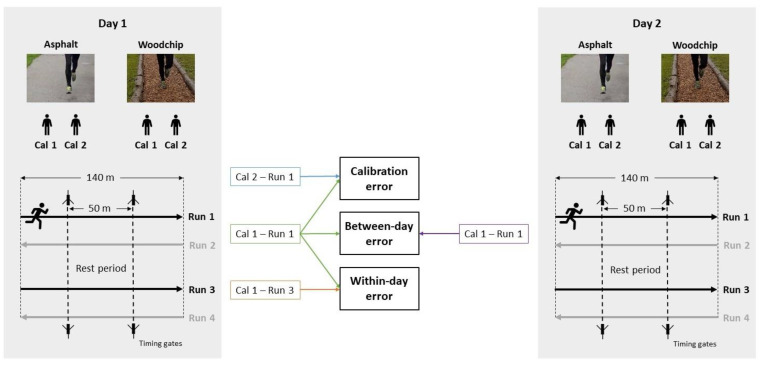
Schematic of experimental protocol and reliability analyses. Note that the calculations of calibration, within-day, and between-day error are only shown for testing day 1 and testing day 2, although the same comparisons were carried out between all five testing days. Runs 2 and 4 (marked in grey) were not included in the analysis. Cal = calibration.

**Figure 2 sensors-24-00871-f002:**
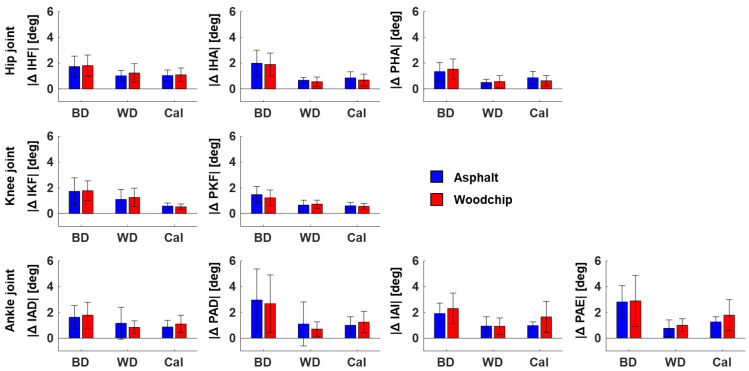
Mean absolute differences (±SD) in Xsens-based lower extremity joint angles between runs on different days (BD), different runs within the same day but processed with the same calibration file (WD) and the same run within the same day but processed with different calibration files (Cal). Blue and red bars show results for runs on asphalt and the woodchip track, respectively. IHF = initial hip flexion, IHA = initial hip ab-/adduction, PHA = peak hip adduction, IKF = initial knee flexion, PKF = peak knee flexion, IAD = initial ankle dorsiflexion, PAD = peak ankle dorsiflexion, IAI = initial ankle inversion, PAE = peak ankle eversion.

**Figure 3 sensors-24-00871-f003:**
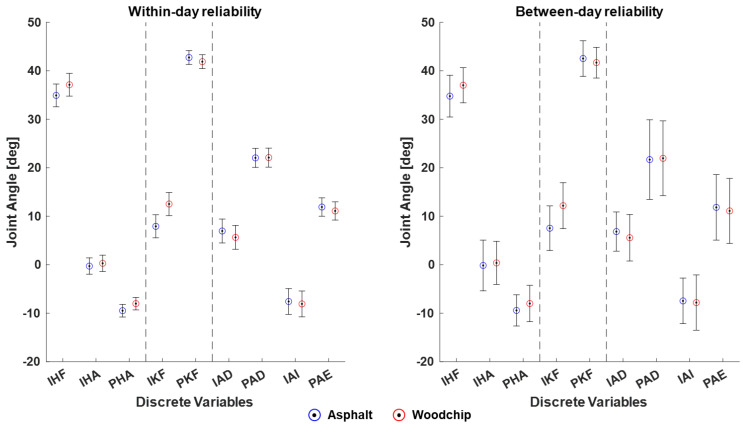
Within-day and between-day reliability of Xsens-based lower extremity joint angles during running. Circles (blue = asphalt, red = woodchip) indicate the grand mean joint angles across runners and testing sessions. Error bars represent the mean angle plus-minus one median minimal detectable change (based on five reliability estimates). IHF = initial hip flexion, IHA = initial hip ab-/adduction, PHA = peak hip adduction, IKF = initial knee flexion, PKF = peak knee flexion, IAD = initial ankle dorsiflexion, PAD = peak ankle dorsiflexion, IAI = initial ankle inversion, PAE = peak ankle eversion.

**Table 1 sensors-24-00871-t001:** ICCs for between-day, within-day and calibration reliability of Xsens-based lower extremity joint angles, as well as running speed and stride frequency during running on an asphalt and woodchip surface. Orange, yellow, light green, and dark green coloring indicate poor, moderate, good, and excellent reliability according to Koo and Li [28].

		Grand Mean ^1^ (deg)		Between-Day Reliability		Within-Day Reliability		Calibration Reliability	
Group	Variable			ICC Median [Range]		ICC Median [Range]		ICC Median [Range]	
		Asphalt	Woodchip	Asphalt	Woodchip	Asphalt	Woodchip	Asphalt	Woodchip
**Hip**	Initial hip flexion (IHF)	34.77	37.02	0.92 [0.89, 0.95]	0.96 [0.83, 0.96]	0.98 [0.95, 0.98]	0.98 [0.91, 0.99]	0.97 [0.96, 0.99]	0.97 [0.96, 0.98]
	Initial hip ab-/adduction (IHA)	−0.15	0.38	0.52 [0.44, 0.68]	0.49 [0.29, 0.62]	0.96 [0.87, 0.98]	0.94 [0.86, 0.99]	0.91 [0.83, 0.94]	0.93 [0.77, 0.99]
	Peak hip adduction (PHA)	−9.42	−8.00	0.83 [0.70, 0.85]	0.74 [0.63, 0.77]	0.97 [0.92, 0.99]	0.96 [0.74, 0.99]	0.89 [0.87, 0.95]	0.96 [0.87, 0.97]
**Knee**	Initial knee flexion (IKF)	7.53	12.18	0.92 [0.90, 0.95]	0.92 [0.89, 0.96]	0.97 [0.96, 0.98]	0.97 [0.95, 0.99]	0.99 [0.99, 0.99]	0.99 [0.99, 1.00]
	Peak knee flexion (PKF)	42.53	41.69	0.82 [0.75, 0.88]	0.89 [0.85, 0.93]	0.96 [0.95, 0.97]	0.96 [0.95, 0.98]	0.97 [0.96, 0.97]	0.98 [0.97, 0.99]
**Ankle**	Initial ankle dorsiflexion (IAD)	6.83	5.56	0.90 [0.82, 0.94]	0.88 [0.87, 0.95]	0.97 [0.52, 0.99]	0.98 [0.90, 0.99]	0.97 [0.96, 0.98]	0.96 [0.91, 0.99]
	Peak ankle dorsiflexion (PAD)	21.68	21.95	0.38 [0.23, 0.52]	0.47 [−0.02, 0.49]	0.93 [0.30, 0.99]	0.97 [0.61, 0.99]	0.88 [0.82, 0.97]	0.91 [0.57, 0.93]
	Initial ankle inversion (IAI)	−7.47	−7.81	0.81 [0.77, 0.90]	0.72 [0.64, 0.85]	0.96 [0.92, 0.98]	0.95 [0.87, 0.98]	0.95 [0.93, 0.97]	0.84 [0.73, 0.93]
	Peak ankle eversion (PAE)	11.83	11.10	0.55 [0.53, 0.72]	0.65 [0.46, 0.72]	0.97 [0.94, 0.98]	0.97 [0.92, 0.98]	0.94 [0.89, 0.96]	0.84 [0.77, 0.94]
**Spatio-** **tem-poral**	Running speed (RS)	3.47	3.38	0.93 [0.84, 0.95]	0.93 [0.76, 0.96]	0.94 [0.84, 0.97]	0.97 [0.90, 0.98]	*not applicable*	
	Stride frequency (SF)	1.40	1.39	0.88 [0.81, 0.88]	0.73 [0.55, 0.87]	0.93 [0.90, 0.95]	0.89 [0.73, 0.95]		

^1^ Grand mean represents the average variable across the first run on each testing day and across all participants.

**Table 2 sensors-24-00871-t002:** Model summary for the association of absolute between-day differences in Xsens-based lower extremity joint angles with the running surface and absolute between-day differences in running speed and stride frequency. Green coloring: These models contained at least one significant coefficient (*p* < 0.05) of between-day differences in the respective joint angle. Grey coloring: These models did not contain a significant coefficient (*p* < 0.05) for the differences between the individual days in the respective joint angle.

Group	Variable	R-Squared Marginal	Between-Day Difference in Running Speed (*b_speed_*) (deg per m/s)	Between-Day Difference in Stride Frequency (*b_SF_*) (deg per 1/s)	Surface (Woodship—Asphalt, *b_Surface_*) (deg)
**Hip**	Initial hip flexion	0.26	4.23 (3.06, 5.40)	2.55 (−6.43, 11.54)	−0.02 (−0.41, 0.36)
	Initial hip ab-/adduction	0.01	−0.69 (−2.18, 0.80)	−1.05 (−13.51, 11.40)	−0.08 (−0.53, 0.37)
	Peak hip abduction	0.02	−0.48 (−1.56, 0.60)	4.89 (−4.20, 13.99)	0.18 (−0.15, 0.49)
**Knee**	Initial knee flexion	0.01	0.15 (−1.25, 1.54)	6.01 (−5.71, 17.73)	0.01 (−0.40, 0.43)
	Peak knee flexion	0.06	0.04 (−0.99, 1.07)	11.32 (2.71, 19.94)	−0.31 (−0.62, −0.00)
**Ankle**	Initial ankle dorsiflexion	0.02	1.03 (−0.46, 2.53)	0.92 (−11.58, 13.41)	0.14 (−0.30, 0.59)
	Peak ankle dorsiflexion	0.12	−0.57 (−3.12, 1.99)	52.10 (30.00, 74.19)	−0.54 (−1.28, 0.19)
	Initial ankle inversion	0.03	−0.60 (−2.25, 1.04)	10.04 (−3.58, 23.66)	0.35 (−0.14, 0.85)
	Peak ankle eversion	0.12	−0.34 (−2.66, 1.99)	41.01 (21.74, 60.28)	−0.12 (−0.82, 0.58)

**Table 3 sensors-24-00871-t003:** ICCs for between-day reliability of Xsens-based lower extremity joint angles during running compared to previously reported ICC between-day reliability estimates of marker-based optical motion capture (OMC) and markerless motion analysis. Orange, yellow, light green, and dark green coloring indicate poor, moderate, good, and excellent reliability according to Koo and Li [28].

Category	Variable	Current Study IMU-Based (Xsens)		Okahisa et al. (2023) [39] OMC (Vicon)		Stoneham et al. (2019) [40] OMC (Vicon)	Moran et al. (2023) Markerless [41] (Theia/Visual3D)
		ICC Median [Range]		ICC		ICC [95% CI]	ICC [95% CI]
		Asphalt	Woodchip	Lab Floor PiG *	Lab Floor CGM2 *	Lab Floor PiG *	Lab Floor
**Hip**	Initial hip flexion	0.92 [0.89, 0.95]	0.96 [0.83, 0.96]	0.80	0.87	0.89 [0.72, 0.96]	0.86 [0.80, 0.91]
	Initial hip ab-/adduction	0.52 [0.44, 0.68]	0.49 [0.29, 0.62]	0.67	0.75	0.81 [0.54, 0.93]	0.75 [0.64, 0.83]
	Peak hip abduction	0.83 [0.70, 0.85]	0.74 [0.63, 0.77]	0.45	0.84	0.69 [0.31, 0.88]	0.91 [0.87, 0.94]
**Knee**	Initial knee flexion	0.92 [0.90, 0.95]	0.92 [0.89, 0.96]	0.74	0.81	0.76 [0.44, 0.91]	0.88 [0.83, 0.92]
	Peak knee flexion	0.82 [0.75, 0.88]	0.89 [0.85, 0.93]	0.74	0.89	0.78 [0.49, 0.92]	0.90 [0.86, 0.94]
**Ankle**	Initial ankle dorsiflexion	0.90 [0.82, 0.94]	0.88 [0.87, 0.95]	0.87	0.93	0.95 [0.86, 0.98]	0.94 [0.90, 0.96]
	Peak ankle dorsiflexion	0.38 [0.23, 0.52]	0.47 [−0.02, 0.49]	0.68	0.86	0.85 [0.63, 0.95]	0.76 [0.66, 0.84]
	Initial ankle inversion	0.81 [0.77, 0.90]	0.72 [0.64, 0.85]	*not available*		0.69 [0.31, 0.88]	0.72 [0.61, 0.81]
	Peak ankle eversion	0.55 [0.53, 0.72]	0.65 [0.46, 0.72]	*not available*		0.74 [0.41, 0.90]	0.83 [0.75, 0.88]

* PiG = Plug-in-Gait model; CGM2 = Conventional Gait Model 2.

## Data Availability

All raw data underlying this study have been uploaded to the Mendeley Data Repository: Mohr, Maurice; Debertin, Daniel; Wargel, Anna; Peer, Lukas; De Michiel, Alessia; van Andel, Steven; Federolf, Peter (2023), “Whole-body kinematic adaptations to running on an unstable, irregular, and compliant surface: Data, code, and Appendix A”, Mendeley Data, V4, https://doi.org/10.17632/j4bck84rpr.5 (accessed on 15 January 2024).

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
