# Peer review of "Reliability of Xsens IMU-Based Lower Extremity Joint Angles during In-Field Running"

_sensors, 2024, doi:10.3390/s24030871_

Round 1

Reviewer 1 Report

Comments and Suggestions for Authors

Reviewer #1: Sensors-2821124

Reliability of Xsens IMU-based lower extremity joint angles during in-field running

This paper assessed the reliability of the Xsens when running on two different surface types (asphalt vs. woodschip). The authors wanted to assess between and within-day reliability for the hip, knee and ankle joints.

The main concerns I have about this paper relate to:

1) Methods

Subjects: The current population for the study consists of individuals who have (ran at least one time per week for the last year, and have not had a lower extremity injury in the past 6 months, that caused for a cease of running for greater than 2 weeks). How is this population classified as runners? Why are there no stipulations, or questionnaire used to tie back to benchmarks or recommendations from ACSM, or other credible sources?

Experimental Protocols: Runners completed four runs along a 140m long straight track with the assigned surfaces totaling 560 total meters per day (surface specific). Did the participants repeat the 140m runs? If so, how long was the rest period between attempts? Did the participants, turn around at the end of 140m and run the other way? Greater elaboration is needed here.

            It was suggested that each participant run at a moderate intensity that they would typically run at for a 60-minute duration. How do the readers know, that the runners could even do this (inclusion criteria never elaborates to this).

Were there any controls for shoes that the runners were allowed to wear? Did they all wear neutral, over or under pronated shoes? Were they running shoes, trail shoes? Was there a baseline assessment on the arch of the foot, or anything along running mechanics prior to the start of the study?

Reviewer 2 Report

Comments and Suggestions for Authors

The introduction presents sufficiently strong arguments that justify the importance of the study. The objectives are clearly established and can help understand the limitations of using Xsens in field measurements. Regarding the methodology, the schematic figure (figure 1) could be improved (the notes for runs 2 and 4 were barely noticeable, apparently only in runs 1 and 3 there was a measurement of running time). The locations where the sensors are positioned on the suit and whether there is a change in size depending on the anthropometry of the participants could be better detailed. Other specific questions are in the attached file. The article is very detailed, the conclusions are consistent with the objectives and results found.

Reviewer 3 Report

Comments and Suggestions for Authors

Movement analysis during dynamic physical activities such as running in natural conditions is very necessary. Understanding what happens while running outside the laboratory is an important element of injury prevention.

Very interesting and innovative work, well conducted by the authors. The work was methodologically well conducted and well-chosen statistic.

One notes for a minor revision:

Line 101 - not only the absence of lower limb injury, but also pain of a different nature, e.g. overload, should be taken into account when qualifying for the test group. It probably was, so it needs to be added.

Round 2

Reviewer 1 Report

Comments and Suggestions for Authors

No further questions or concerns